

# Hsa_circ_0001615 downregulation inhibits esophageal cancer development through miR-142-5p/β-catenin

Yukai Dai[1,2,*], Qizhong Xu[1,*], Manqi Xia[1], Caimin Chen[1], Xinming Xiong[1,2], Xin Yang[1,2] and Wei Wang[1,2]

[1] The Second Clinical College of Guangzhou Medical University, Guangzhou, China
[2] Department of Thoracic Surgery, The Second Affiliated Hospital of Guangzhou Medical University, Guangzhou, China
* These authors contributed equally to this work.

## ABSTRACT

**Background:** Recent studies have found that circular RNAs (circRNAs) play important roles in tumorigenesis. This study aimed to determine the function and potential mechanisms of hsa_circ_0001615 in esophageal cancer.

**Methods:** Quantitative real-time reverse transcription polymerase chain reaction (qRT-PCR) was used to validate the expression of hsa_circ_0001615 and miR-142-5p. Subsequently, 3-(4,5-dimethylthiazol-2-yl)-5-(3-carboxymethoxyphenyl)-2-(4-sulfophenyl)-2H-tetrazolium salt, flow cytometry, clone formation, and transwell assays were used to assess the function of hsa_circ_0001615. Furthermore, qRT-PCR and Western blot analysis were used to verify cyclin D1, Bcl-2 associated X, B-cell lymphoma/leukemia gene-2, and β-catenin levels. Circular RNA Interactome was used to estimate the binding site between hsa_circ_0001615 and miR-142-5p. Additionally, dual-luciferase reporter assays were used to determine whether miR-142-5p was a direct target of hsa_circ_0001615. Pearson correlation analysis was used to explore the relationship between miR-142-5p and hsa_circ_0001615.

**Results:** In esophageal cancer, the expressions of hsa_circ_0001615 and miR-142-5p were increased and decreased, respectively. Hsa_circ_0001615 inhibition significantly reduced the proliferation, migration, and invasion but increased the apoptosis of esophageal cancer cells. Additionally, hsa_circ_0001615 knockdown increased miR-142-5p expression but decreased β-catenin expression. MiR-142-5p was a direct target of hsa_circ_0001615.

**Conclusion:** Hsa_circ_0001615 knockdown could mediate antitumor effects through the miR-142-5p/β-catenin pathway.

Corresponding author
Wei Wang,
2011682026@gzhmu.edu.cn

## INTRODUCTION

Esophageal cancer is the sixth most common cause of cancer-related deaths and the eighth most common cancer worldwide, with particularly high incidence in Western Asia (*Bray et al., 2018*). Despite therapeutic advancements in recent decades, esophageal cancer still has very poor prognosis, with an overall 5-year survival rate of approximately 10% (*Huang & Yu, 2018*). Because of the lack of obvious pathogenic features and early diagnostic
markers, patients with esophageal cancer are typically diagnosed in late stages, limiting the effectiveness and possibilities for treatment. Thus, the underlying mechanisms of esophageal cancer need to be elucidated, and promising therapeutic targets need to be identified.

Circular RNAs (circRNAs) are noncoding RNAs that are endogenously and stably expressed in mammals (*Memczak et al., 2013*). They lack 5′-3′ polarity and polyA tails because they are covalently closed loops (*Zhang et al., 2018*). CircRNAs regulate gene expression or protein function by acting as microRNA or protein sponges (*Kristensen et al., 2019*). They have been recognized as novel biomarkers for cancers because of their high tissue specificity and stability (*Kristensen et al., 2022*). For example, hsa_circ_0032746 has been found to regulate the oncogenic process of esophageal squamous cell carcinoma by modulating the miR-4270/MCM3 axis (*Shrestha et al., 2024*). Circ_ZNF778_006 promotes the progression of esophageal squamous cell carcinoma by upregulating HIF-1α expression *via* sponging miR-18b-5p (*Si et al., 2023*). Although many studies have investigated circRNAs in esophageal cancer, the mechanisms of most circRNAs in esophageal cancer remain unknown.

Hsa_circ_0001615, also called circPHIP, has been reported to promote the progression of oral squamous cell carcinoma (OSCC) by sponging miR-142-5p and regulating the expression of PHIP and ACTN4 (*Su et al., 2021*). However, no studies have reported its significance in esophageal cancer. In the present study, the expression of hsa_circ_0001615 in esophageal cancer tissue and cells was determined. The impact and possible underlying mechanisms of hsa_circ_0001615 in esophageal cancer were evaluated through siRNA knockdown of hsa_circ_0001615. The results will provide novel insights for identifying tumor markers or molecular targets for esophageal cancer treatment.

## MATERIALS AND METHODS

### Cell culture

The 293T cell line and human esophageal cancer cells (TE-1, KYSE520, and ECA-109) were obtained from Guangzhou Cellcook Cell Biotechnology, Ltd., (Guangzhou, China). Normal esophageal epithelial cells (HEEC) were purchased from ScienCell (San Diego, CA, USA). All cells were cultured in Dulbecco's Modified Eagle Medium (DMEM) + 10% fetal bovine serum (FBS) (Thermo Fisher Scientific, Waltham, MA, USA) and maintained at 37 °C under 5% $CO_2$.

### RNA Interference of Hsa_circ_0001615

Small interfering RNAs (siRNAs) were acquired from GenePharma (Shanghai, China). The interference sequences are as follows: si-hsa_circ_0001615 sense, 5′-ACUUCACA GUAAAUAUCCGUC-3′; antisense, 5′-ACUUCACAGUAAAUAUCC GUC-3′; negative control (NC) siRNA sense, 5′-UUCUCCGAACGUGUCACGUTT-3′; antisense, 5′-ACGUGACACGUUCGGA GAATT-3′. The NC was a random siRNA that was also provided by GenePharma. Transfection was carried out using Lipofectamine 2000 (Thermo Fisher Scientific, Waltham, MA, USA) in accordance with the manufacturer's

protocols. Briefly, the cells were seeded into six-well plates ($1 \times 10^3$ cells per well) and then treated with 200 pmol of siRNA for 48 h.

## 3-(4,5-Dimethylthiazol-2-yl)-5-(3-carboxymethoxyphenyl)-2-(4-sulfophenyl)-2H-tetrazolium Salt (MTS) and cloning assays for cell proliferation

After 48-h transfection with siRNAs, the cells were digested with trypsin, and $1 \times 10^4$ cells in 100 µL were plated in a 96-well plate. About 10 µL of MTS reagent (Promega, Madison, WI, USA) was added. After incubating cells with MTS reagents at 37 °C for 4 h, the absorbance at 490 nm was measured with a spectrophotometer (Diatek, Inc., San Diego, CA, USA). Each condition was measured in three biological replicates.

The proliferation of ECA-109 and KYSE520 cells transfected with siRNAs was also assessed by cloning assay. A total of four hundred cells were seeded into a six-well plate after 48-h transfection with siRNAs. One week after seeding, the cells were rinsed twice with PBS and then fixed in 4% paraformaldehyde for 10 min. Crystal violet was used to visualize the cells. The number of clones was calculated using ImageJ software. The relative clone formation ability was calculated as follows: (clone number/mean clone number of the NC group) × 100%.

### Transwell assay for cell migration and invasion

Approximately $1 \times 10^5$ cells were seeded in 100 µL of serum-free DMEM in the top chamber of each transwell for the migration assay. Meanwhile, $2 \times 10^5$ cells were seeded in 100 µL of serum-free DMEM inside an invasion chamber coated with Matrigel for the invasion assay. Approximately 600 µL of DMEM containing 10% FBS was delivered to the lower compartment of each transwell. After 24 h of migration and 48 h of invasion incubation, the cells kept in the upper chambers were cleaned with cotton swabs. The transwells were fixed in 4% paraformaldehyde. Crystal violet was used to visualize the cells. The migrated and invaded cells were imaged using an inverted microscope (Optec, Chongqing, China).

### Western blot analysis

About 120 µL of RIPA was added to lyse ECA-109 and KYSE520 cells transfected with siRNAs for 48 h. The cell lysates were transferred into tubes and then ultrasonicated (Ningbo Xinzhi Medicine UP-250; Ningbo Xinzhi Medicine, Ningbo, China). The proteins from the supernatant were extracted for bicinchoninic acid quantification, and 30 µg of total protein was subjected to sodium dodecyl-sulfate polyacrylamide gel electrophoresis followed by electrotransfer with PVDF membrane. The primary antibodies were incubated overnight at 4 °C after blocking with skim milk (5%). The details of the antibodies used in this study are shown in Table 1. The secondary antibodies conjugated with HRP (Thermo Fisher Scientific, Waltham, MA, USA) were incubated for 1 h. GAPDH was used as an internal reference and normalized to the control group. The membranes were finally developed using ECL reagents (Thermo Fisher Scientific, Waltham, MA, USA). The band intensities were analyzed using ImageJ software (NIH Image, Bethesda, MA, USA).

**Table 1 List of antibodies.**

| Antibody | Company | Product number | Dilution |
| --- | --- | --- | --- |
| β-catenin | Abcam | ab32572 | 1/10,000 |
| CyclinD1 | Abcam | ab134175 | 1/1,000 |
| Bax | Proteintech | 50599-2-Ig | 1/8,000 |
| Bcl2 | Cell signaling technology | 3498 | 1/1,000 |
| GAPDH | Proteintech | 60004-1-Ig | 1/10,000 |

## Collection of specimens

Esophageal cancer tissues (30 pairs) and adjacent nontumor esophageal tissues were collected from patients who underwent esophageal resection at the Second Affiliated Hospital of Guangzhou Medical University Hospital. This study was approved by the Ethics Committee of the Second Affiliated Hospital of Guangzhou Medical University (Clearance certificate number: 2021-ks-KY-07). All patients who participated in the present study provided written informed consent. The clinicopathological characteristics of 30 patients with esophageal cancer are shown in Table 2.

## Quantitative real-time reverse transcription polymerase chain reaction (qRT-PCR)

Total RNA was isolated using TRIzol (MRC TR118-500). Generally, good quality RNA has an A260/A280 ratio of 1.8–2.1. About 2 μL of 40 μg/mL of each RNA sample was transcribed into cDNA using M-MLV Reverse Transcriptase (Promega M1705; Promega, Madison, WI, USA) for circRNA and mRNA qRT-PCR. For miRNA qRT-PCR, the mature miRNA was reverse transcribed using the miRNA-specific stem-loop primer. PCR was conducted using the GoTaq® qPCR Master Mix (Promega A6002; Promega, Madison, WI, USA) on ABI 7500. The relative expression of hsa_circ_0001615 and mRNAs was normalized to GAPDH, whereas that of miRNAs was normalized to U6, respectively. The relative expression was calculated based on the $2^{-\Delta\Delta CT}$ method (*Livak & Schmittgen, 2001*) and normalized to the control group. The primers are shown in Table 3. Specifically, convergent and divergent primers were used to quantify circRNAs. Both primers were used for PCR with total ECA-109 RNA with and without RNase-R treatment. Then, the PCR products were subjected to gel electrophoresis followed by Sanger sequencing with divergent products to confirm the circularity of the amplified RNA.

## Flow cytometry for apoptosis

Flow cytometry staining for APC Annexin V (KeyGEN BioTECH KGA1022; KeyGEN BioTECH, Jiangsu, China) and 7-AAD (eBioscience 00-6993-50; eBioscience, San Diego, CA, USA) was used to measure the apoptosis phenotypes of cells. For each sample, $1 \times 10^5$ cells were mixed with Annexin V (5 μL) and then added with 7-AAD. After 15 min of incubation, the apoptosis rates were analyzed using flow cytometry (Beckman Cytoflex; Beckman, Brea, CA, USA). The apoptosis rate was the sum of the early and late apoptosis rates.

Table 2 The clinicopathological characteristics of 30 esophageal cancer patients.

| Characteristics | Case number |
| --- | --- |
| **Sex** | |
| Female | 7 (23.3%) |
| Male | 23 (76.7%) |
| **Age** | |
| <60 | 15 (50.0%) |
| ≥60 | 15 (50.0%) |
| **Tumor location** | |
| Upper | 7 (23.3%) |
| Middle | 6 (20.0%) |
| Lower | 17 (56.7%) |
| **Depth of invasion (pT)** | |
| Adjacents structures | 2 (6.6%) |
| Adwentitia | 15 (50%) |
| Muscularis mucosa | 1 (3.3%) |
| Muscularis propria | 11 (36.6%) |
| Submucosa | 1 (3.3%) |
| **Lymph node metastasis (pN)**[*] | |
| 0 | 19 (63.3%) |
| 1 | 3 (10.0%) |
| 2 | 2 (6.6%) |
| 3 | 2 (6.6%) |
| 4 | 2 (6.6%) |
| 5 | 2 (6.6%) |
| **Pathological stage(pStage)**[*] | |
| IA | 1 (3.3%) |
| IB | 1 (3.3%) |
| IC | 7 (23.3%) |
| IIA | 2 (6.6%) |
| IIB | 8 (26.6%) |
| IIIA | 2 (6.6%) |
| IIIB | 8 (26.6%) |
| IVA | 1 (3.3%) |
| **Tumor differentiation** | |
| G1 | 13 (43.3%) |
| G2 | 8 (26.6%) |
| G3 | 9 (30%) |

**Note:**

[*] Lymph node metastasis (pN) and pathological stages (pStage) were determined according to the Unio Internationalis Contra Cancrum (UICC) 8[th] edition.

## Dual-luciferase reporter assays

The wild-type hsa_circ_0001615-wt or mutant hsa_circ_0001615-mut sequences were ligated into the pmirGLO vector (Promega, Madison, WI, USA) for dual-luciferase
**Table 3 List of primers.**

| Gene name | Sequences (5′–3′) | Length (bp) | Product size (bp) | GC % | Hairpin |
|---|---|---|---|---|---|
| hsa_circ_0001615- diver-F | TGCAGTGTATCAGCACATGAA | 21 | 144 | 42.9 | Found |
| hsa_circ_0001615- diver-R | ATTTGCAGCAAGTGATCAGG | 20 | | 45.0 | None |
| hsa_circ_0001615- conver-F | GTGGAAAGGATCTGCTCTGG | 20 | 146 | 55.0 | None |
| hsa_circ_0001615-conver-R | CAGTTGGAACAAGTCGCTCA | 20 | | 50.0 | Found |
| hsa-miR-142-5p-RT | CTCAACTGGTGTCGTGGAGTCGGCAATTCAGTTGAGAGTAGT | 42 | 57 | 50.0 | Found |
| hsa-miR-142-5p-F | GCCGAGCATAAAGTAGAAAGC | 21 | 63 | 47.6 | None |
| Universe-R | CTCAACTGGTGTCGTGGA | 18 | | 55.6 | None |
| U6-F | CTCGCTTCGGCAGCACA | 17 | 96 | 64.7 | Found |
| U6-R | AACGCTTCACGAATTTGCGT | 20 | | 45.0 | Found |
| BAX-F | CCCGAGAGGTCTTTTTCCGAG | 21 | 109 | 57.1 | None |
| BAX-R | GCCTTGAGCACCAGTTTGC | 19 | | 57.9 | None |
| BCL2-F | GGTGGGGTCATGTGTGTGG | 19 | 89 | 63.2 | None |
| BCL2-R | CGGTTCAGGTACTCAGTCATCC | 22 | | 54.5 | None |
| CyclinD1-F | GCTGCGAAGTGGAAACCATC | 20 | 135 | 55.0 | None |
| CyclinD1-R | CCTCCTTCTGCACACATTTGAA | 22 | | 45.5 | None |
| GAPDH-F | GAGTCAACGGATTTGGTCGT | 20 | 131 | 50.0 | None |
| GAPDH-R | GACAAGCTTCCCGTTCTCAG | 20 | | 55.0 | None |

reporter assays. Thereafter, these constructs were co-transfected into 293T cells for 48 h using Lipofectamine 2000 in combination with miR-142-5p and NC mimics.

The luciferase reporter assay was performed using the dual-luciferase reporter assay kit (Promega E1910; Promega, Madison, WI, USA) in accordance with the manufacturer's instructions. The results were expressed as the relative firefly luciferase activity (*i.e.*, ratio of firefly luciferase activity to Renilla luciferase activity) and normalized to the hsa_circ_0001615-wt+NC mimics group.

## Statistical analysis

Data analysis was performed using GraphPad Prism 6 (GraphPad Software). All data were presented as the mean ± standard deviation. Two independent or paired groups were compared using Student's *t*-test or Wilcoxon signed rank test. One-way or two-way analysis of variance with the Bonferroni *post hoc* test was used for multiple group analysis. Correlations between hsa_circ_0001615 and miR-142-5p were analyzed using the Pearson correlation test. The following probability symbols were used in all figures: ns (not statistically significant), $*P < 0.05$, $**P < 0.01$, $***P < 0.001$, and $****P < 0.0001$.

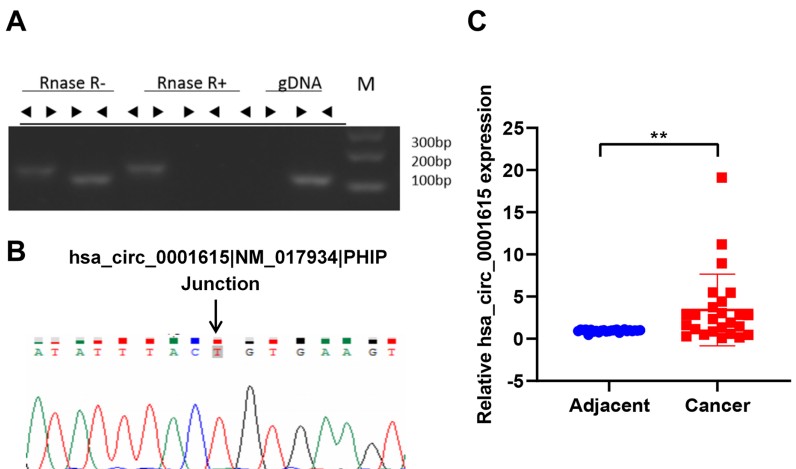

**Figure 1** **Hsa_circ_0001615 is elevated in esophageal cancer.** (A) RNase R treatment of PCR amplicons from convergent primers and divergent primers of hsa_circ_0001615, illuminating circularity of hsa_circ_0001615; (B) Sanger sequencing of PCR amplicons from divergent primers confirming the backsplice junction of hsa_circ_0001615; (C) qRT-PCR detection of hsa_circ_0001615 in cancer tissue and adjoining noncancerous tissue. **$P < 0.01$.

# RESULTS

## Hsa_circ_0001615 expression was elevated in esophageal cancer

To identify the circular form of hsa_circ_001615, convergent and divergent primers (listed in Table 3) were designed for PCR in both cDNA and gDNA. Next, the PCR products of cDNA were subjected to RNase R treatment. Divergent primers could amplify circRNA in cDNA but not in gDNA by PCR. By contrast, convergent primers could amplify products in both cDNA and gDNA. In addition, the amplification products of divergent primers in cDNA were resistant to RNase R treatment (Fig. 1A). These results demonstrated the cyclic structure and stability of hsa_circ_001615. Sanger sequencing was performed to further validate the circularity. The sequencing data confirmed the presence of the back-splice junction on the amplicon as anticipated (Fig. 1B). All results confirmed the specificity of the divergent primers and the circular nature of hsa_circ_001615.

The expression of hsa_circ_0001615 was quantified in esophageal cancer and adjacent noncancerous tissues (30 pairs) by qRT-PCR using the divergent primers. The data confirmed that the expression of hsa_circ_0001615 was notably greater in cancer tissue than in adjacent noncancerous tissue (Fig. 1C). The elevated expression of hsa_circ_0001615 indicated that hsa_circ_0001615 likely has a tumor-promoting function in esophageal cancer.

## Cell proliferation was reduced in Hsa_circ_0001615 knockdown cells

The endogenous expression of hsa_circ_0001615 was quantified in three esophageal cancer cell lines (ECA-109, TE-1, KYSE520) and normal esophageal cells (HEEC). The relative expression of hsa_circ_0001615 was higher in the three esophageal cancer cell lines than in normal esophageal cells. Moreover, it was higher in ECA-109 and KYSE520 cells than in TE-1 cells (Fig. 2A).

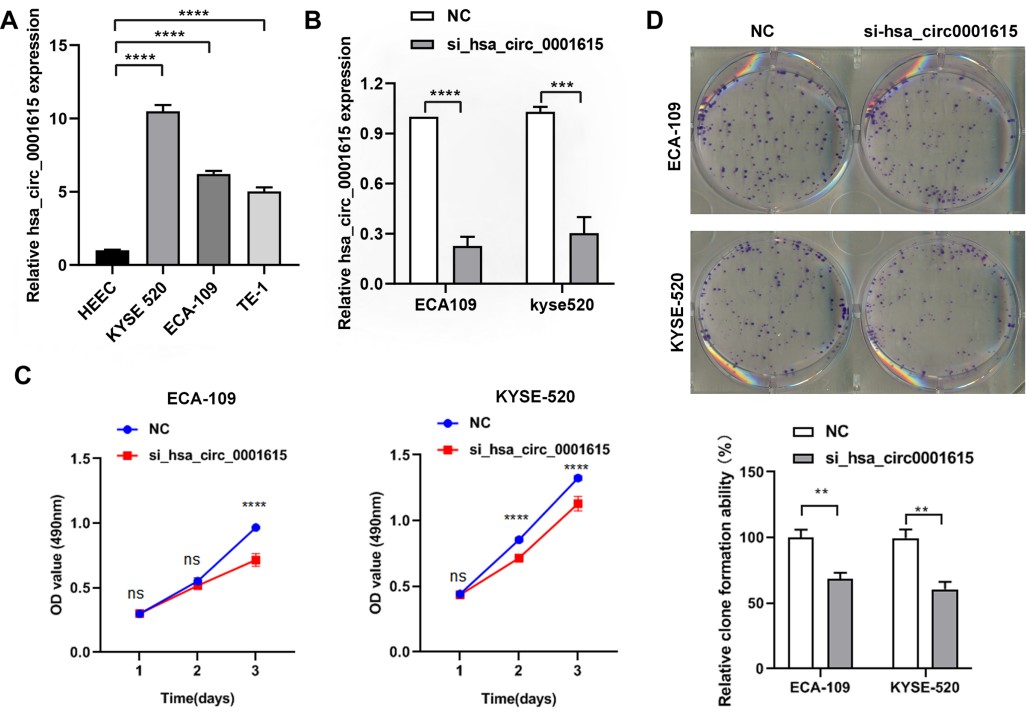

**Figure 2 Reduced cell proliferation in hsa_circ_0001615 knockdown cells.** (A) qRT-PCR detection of hsa_circ_0001615 in esophageal cells HEEC, TE-1, ECA-109, and KYSE520; (B) siRNA interfering hsa_circ_0001615 expression markedly repressed hsa_circ_0001615 expression in ECA-109 and KYSE520 cells; (C) MTS assays showed a decreased of cell proliferation in hsa_circ_0001615 knockdown ECA-109 and KYSE520; (D) clone formation assay showed a decreased of clone formation in hsa_circ_0001615 knockdown ECA-109 and KYSE520. ns (not statistically significant), $^{**}P < 0.01$, $^{***}P < 0.001$, and $^{****}P < 0.0001$.

ECA-109 and KYSE520 cell lines were used to transfect hsa_circ_0001615 siRNA to gain a better understanding of the functions of hsa_circ_0001615. The relative expression of hsa_circ_0001615 was significantly reduced after transfection with siRNAs for 48 h (Fig. 2B).

Cell proliferation was significantly impaired when hsa_circ_0001615 was knocked down in ECA-109 and KYSE520 cells, as demonstrated by MTS assays (Fig. 2C). Specifically, ECA-109 cells showed reduced cell viability on day 3, whereas KYSE520 cells showed reduced cell proliferation on day 2 (Fig. 2C). Cloning assay was also performed to confirm the growth defect in hsa_circ_0001615 knockdown cell lines. The number of clone formation was substantially decreased when hsa_circ_0001615 was knocked down (Fig. 2D). Collectively, these results indicated that reduced hsa_circ_0001615 expression decreased cell proliferation and colony formation in ECA-109 and KYSE520 cell lines.

## Cell apoptosis was induced in Hsa_circ_0001615 knockdown cells

Apoptosis-related genes and proteins were detected to investigate the influence of hsa_circ_0001615 knockdown on esophageal cancer cell apoptosis. qRT-PCR was performed to determine the apoptotic impact of hsa_circ_0001615 in ECA-109 and KYSE520 cell lines. Compared with the control group in ECA-109, the expression of Bcl-2

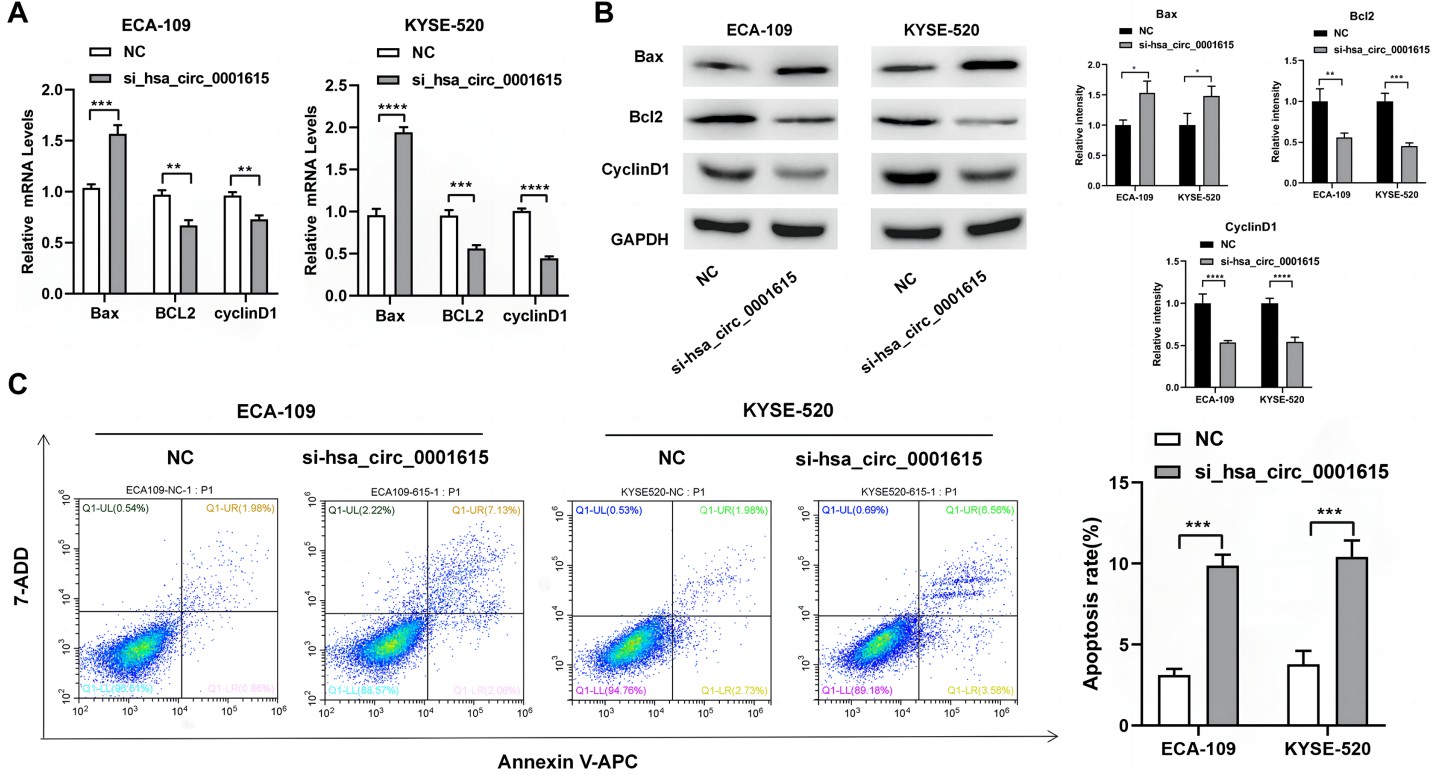

**Figure 3 Promoted apoptosis in hsa_circ_0001615 knockdown cells.** (A) Expressions of Bax, Bcl2, and Cyclin D1 in ECA-109 and KYSE520 cells detected by qRT-PCR, before and after hsa_circ_0001615 was knockdown; (B) expressions of Bcl2, Cyclin D1, and Bax in in ECA-109 and KYSE-520 cells detected by Western blot, before and after hsa_circ_0001615 was knockdown; (C) flow cytometry results showed increased in apoptosis when hsa_circ_0001615 was knocked down. $^{*}P < 0.05$, $^{**}P < 0.01$, $^{***}P < 0.001$, and $^{****}P < 0.0001$.

associated X (Bax) was increased, whereas those of B-cell lymphoma/leukemia gene-2 (Bcl-2) and cyclin D1 were decreased in the hsa_circ_0001615 knockdown group. Similar changes were observed in KYSE-520 cells (Fig. 3A). Western blot analysis was used to evaluate the protein expressions of Bax, Bcl-2, and cyclin D1 (Fig. 3B). After calculating the gray value of the band relative to the expression of the internal reference, the levels were standardized to those of the NC group. Subsequently, statistical analysis was conducted. After knocking down hsa_circ_0001615 in ECA-109 and KYSE520 cell lines, the relative expressions of Bcl-2 and cyclin D1 were decreased, whereas that of Bax was increased.

Flow cytometry analysis demonstrated that hsa_circ_0001615 knockdown cell lines also showed increased apoptosis (Fig. 3C). The percentage of apoptotic cells in ECA-109 and KYSE520 cells after siRNA transfection was measured by staining for Annexin V and 7-AAD. Collectively, the results indicated that reduced hsa_circ_0001615 expression induced cell apoptosis in the ECA-109 and KYSE520 cell lines.

## Cell migration and invasion were repressed in Hsa_circ_0001615 knockdown cells

Transwell assays were performed on hsa_circ_0001615 knockdown cell lines to gain insights into the migratory and invasive behavior of cancer cells. Interestingly, when

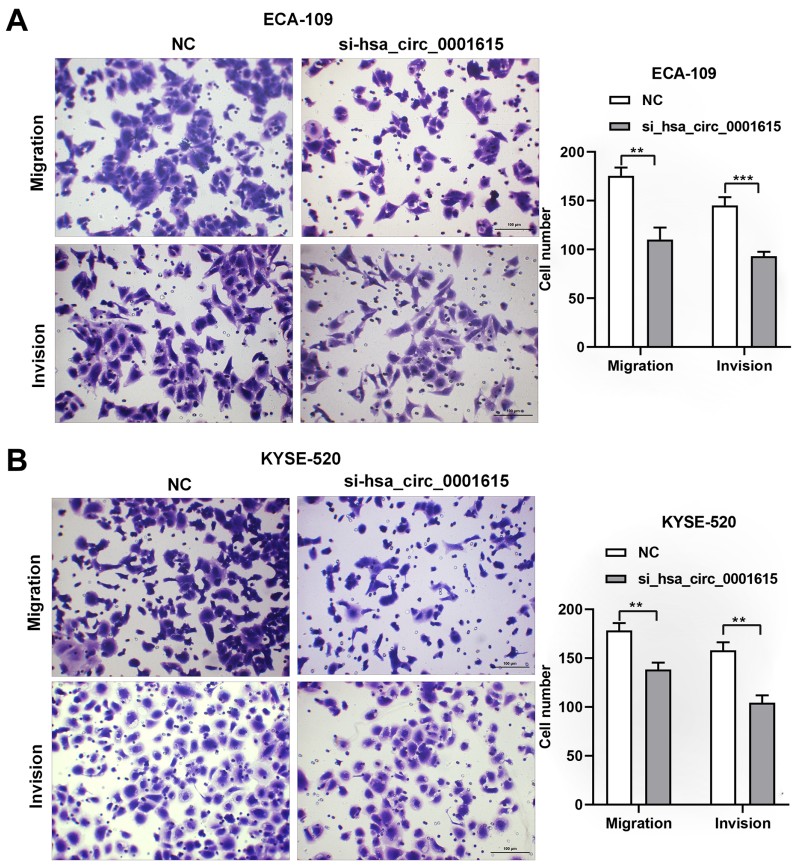

**Figure 4 Migration and Invasion of ECA-109 and KYSE520 cells after hsa_circ_0001615 was Knockdown.** (A) Transwell assays in ECA-109 with hsa_circ_0001615 knockdown showed a decreased number of migration and invasion cells; (B) transwell assays in KYSE-520 with hsa_circ_0001615 knockdown showed a decreased number of migration and invation cells. $**P < 0.01$, and $***P < 0.001$.

hsa_circ_0001615 expression was suppressed, the number of migrating and invading cells was obviously decreased in ECA-109 (Fig. 4A) and KYSE-520 (Fig. 4B) cell lines. Therefore, reduced hsa_circ_0001615 expression decreased the migration and invasion of KYSE520 and ECA-109 cells compared with the control.

## Altered expressions of β-catenin proteins revealed the potential underlying mechanism of Hsa_circ_0001615

To investigate whether hsa_circ_0001615 serves as an miRNA sponge, Circular RNA Interactome was used to predict the downstream miRNAs of hsa_circ_0001615 (*Dudekula et al., 2016*). Among the miRNAs, miR-142-5p has been found to be a direct target of hsa_circ_0001615 in OSCC (*Su et al., 2021*) and a tumor suppressor in esophageal cancer (*Feng et al., 2024*; *Huang et al., 2019*). Therefore, it was focused in this study. Similar to a previous study (*Su et al., 2021*), dual-luciferase reporter assays showed that the luciferase activity of the reporter gene containing the hsa_circ_0001615-wt sequence was significantly reduced, whereas that of the reporter gene containing the hsa_circ_0001615-mut sequence remained unchanged (Fig. 5B). MiR-142-5p expression was quantified in

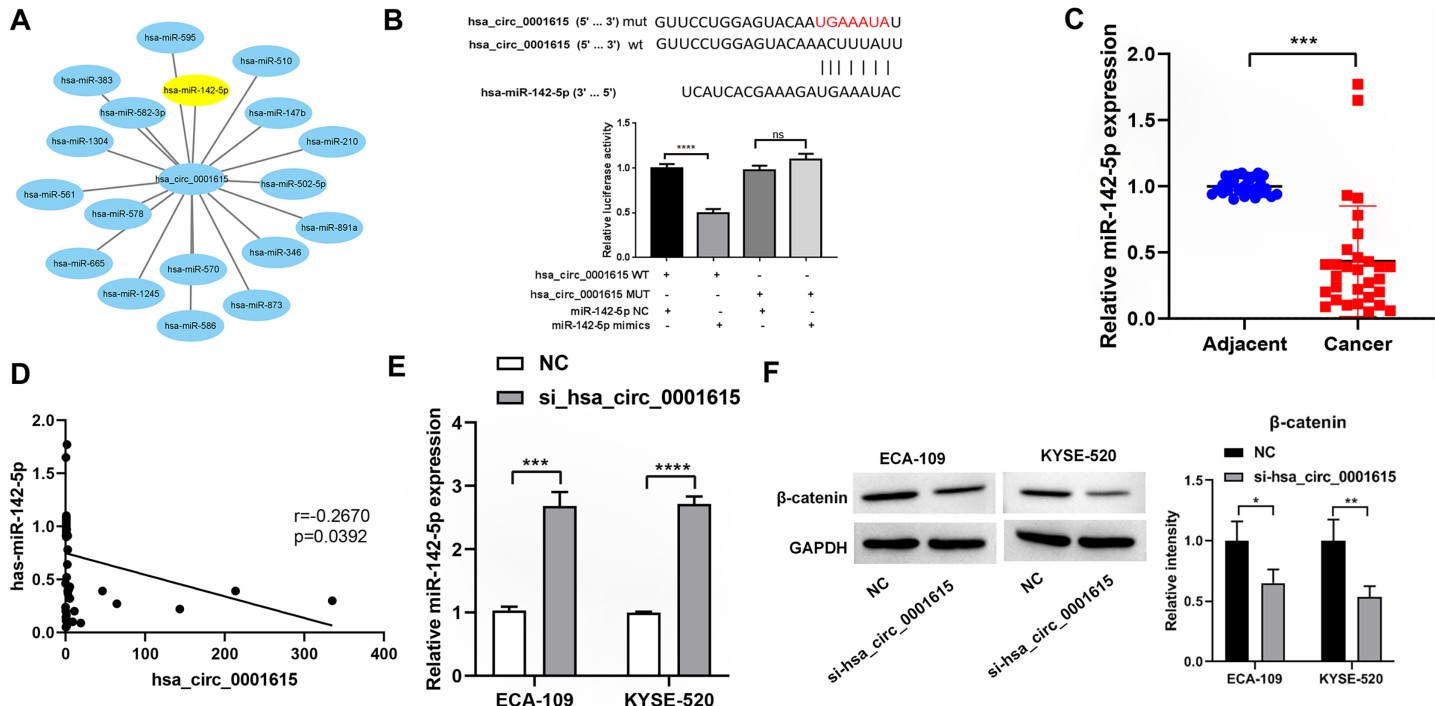

**Figure 5 Hsa_circ_0001615 regulates miR-142-5p through direct binding.** (A) All miRNAs that were predicted to have a targeting relationship with hsa_circ_0001615; (B) Dual luciferase reporter assay to detect the binding site between hsa_circ_0001615 and miR-142-5p; (C) qRT-PCR detection of miR-142-5p in cancer tissue and adjoining noncancerous tissue ($n$ = 30); (D) relationship between hsa_circ_0001615 and miR-142-5p in esophageal tumor tissue ($n$ = 30); (E) miR-142-5p expression detected by qRT-PCR in KYSE520 and ECA-109 cells, before and after hsa_circ_0001615 was knockdown; (F) expressions of β-catenin in KYSE520 and ECA-109 cells detected by Western blot, before and after hsa_circ_0001615 was knockdown. ns (not statistically significant), $^*P < 0.05$, $^{**}P < 0.01$, $^{***}P < 0.001$, and $^{****}P < 0.0001$.

cancerous and adjacent noncancerous tissues from 30 pairs of esophageal cancer samples. qRT-PCR analysis confirmed that noncancerous tissues had higher miR-142-5p levels compared with cancerous tissues (Fig. 5C). The correlation between miR-142-5p and hsa_circ_0001615 expression was analyzed, and the results showed that hsa_circ_0001615 negatively regulates miR-142-5p (Fig. 5D). The qRT-PCR data showed that the expression of miR-142-5p increased when the expression of hsa_circ_0001615 was reduced in ECA-109 and KYSE520 cell lines (Fig. 5E). These results suggested that hsa_circ_0001615 can be a pro-oncogene in esophageal cancer by sequestering miR-142-5p and inhibiting its expression.

Previous studies showed that miR-142-5p regulates the Wnt/β-catenin pathway in cancer (*Ke et al., 2021*; *Wang et al., 2020*). To verify whether β-catenin expression is affected by hsa_circ_0001615 expression, the protein expression of β-catenin in cells was explored by knocking down hsa_circ_0001615. The results showed that the protein expression of β-catenin decreased after knocking down hsa_circ_0001615 in ECA-109 and KYSE520 cell lines (Fig. 5F). Thus, the inhibition of hsa_circ_0001615 regulates β-catenin probably by binding to miR-142-5p directly. This may lead to a decrease in cell proliferation, clone formation, cell migration, and cell invasion while increasing cell apoptosis.

## DISCUSSION

The expression of hsa_circ_0001615 was reported to be elevated in OSCC (*Su et al., 2021*). As a novel cirRNA, the expression of hsa_circ_0001615 in esophageal tumors has never been reported. To the best of our knowledge, the present study is the first to characterize hsa_circ_0001615 expression in esophageal tumor tissues and adjacent nontumor tissues. The data revealed that the expression of hsa_circ_0001615 was elevated in esophageal tumors. By reporting the expression level in patients, our study of hsa_circ_0001615 provides clinical significance.

Bcl-2 family proteins were considered to be key regulators of apoptosis, with Bax being a proapoptotic gene and Bcl-2 being an antiapoptotic gene (*Hata, Engelman & Faber, 2015*). The polymorphisms of Bcl-2 were significantly associated with the risk of developing esophageal cancer (*Jain et al., 2007*). Serving as a prognostic biomarker, Bax polymorphisms affect Bax gene expression, esophageal tumor progression, and metastasis (*Sun, Wei & Li, 2018*). Cyclin D1 controls the cell cycle entry from the G1 to the S phase and is overexpressed in many human cancers, including esophageal cancer (*Samejima et al., 1999*). The overexpression of cyclin D1 is often associated with the worst clinical outcomes and prognosis in patients with esophageal cancer (*Wang et al., 2017*). The present study confirmed that hsa_circ_0001615 downregulation decreased the protein expression of Bcl-2 and cyclin D1 and increased the expression of Bax, potentially explaining the apoptosis-promoting phenotype of hsa_circ_0001615 knockdown. Our results indicated that hsa_circ_0001615 could regulate cellular functions in a complex network in esophageal cancer.

CircRNAs are known to act as miRNA sponges (*Meng et al., 2023*). In the present study, a direct binding site between hsa_circ_0001615 and miR-142-5p was identified, which is consistent with previous reports (*Su et al., 2021*). This finding suggests that hsa_circ_0001615 can act as a endogenous competitor for miR-142-5p. Moreover, the expression of miR-142-5p was diminished in esophageal tumors compared to adjacent nontumor tissues, and hsa_circ_0001615 negatively regulated miR-142-5p. Recent studies have reported the reverse correlation of miRNA levels with the expression of target circRNA, indicating that miRNA expression is regulated by circRNAs (*Han et al., 2020*). Singh et al. found that circRNAs can regulate miRNA expression by (1) recruiting or altering the availability of Pol II/transcription factors to directly regulate miRNA transcription; (2) sequestering miRNAs and RNA binding proteins, thereby regulating the expression of genes involved in miRNA transcription and processing and indirectly regulating mRNA expression; and (3) inhibiting miRNA expression by degrading miRNAs through circRNAs (*Singh, Sinha & Panda, 2023*). Further studies are needed to explore how hsa_circ_0001615 regulates the expression of miR-142-5p.

The Wnt/β-catenin pathway is a highly conserved signaling pathway that regulates cell migration, cell invasion, and immune cell infiltration in the tumor microenvironment (*Zhang & Wang, 2020*). The activation of the β-catenin signaling pathway enhances esophageal cancer tumorigenesis (*Gao et al., 2022*). MiR-142-5p was found to regulate the Wnt/β-catenin pathway in different cancers. For example, miR-142-5p attenuated gastric

cancer cell migration and invasion, at least partially, by inactivating the canonical Wnt/β-catenin signaling pathway (*Yan et al., 2019*). The miR-142-5p-mediated activation of the Wnt/β-catenin signaling pathway was found to promote glioma cell stemness (*Wang et al., 2020*) and cervical cancer progression (*Ke et al., 2021*). However, the regulatory role of miR-142-5p in the Wnt/β-catenin signaling pathway in esophageal cancer remains to be elucidated. We hypothesized that hsa_circ_0001615 could regulate the Wnt/β-catenin signaling pathway by adsorbing miR-142-5p in esophageal cancer. The present study demonstrated that hsa_circ_0001615 knockdown reduced β-catenin expression.

Despite our findings, this study has some limitations. The sample size in this study was very small. A larger sample size will help determine the prognostic role of hsa_circ_0001615, improving the clinical relevance and accuracy of our data. Furthermore, additional cell and animal experiments, as well as rescue experiments, are needed to confirm the mechanism of hsa_circ_0001615 in esophageal cancer.

## CONCLUSION

This study showed the antitumor effect of hsa_circ_0001615 knockdown in esophageal cancer cells, which may be achieved through a miR-142-5p/β-catenin-mediated mechanism. Specifically, it inhibits cell proliferation, migration, and invasion while promoting apoptosis, providing a novel potential target for esophageal cancer.

### Funding
This research was supported By the Open Laboratory Project of College Students at Guangzhou Medical University (2020) and the General guidance project of Health Science and Technology in Guangzhou (20201A011087). The funders had no role in study design, data collection and analysis, decision to publish, or preparation of the manuscript.

### Grant Disclosures
The following grant information was disclosed by the authors:
Open Laboratory Project of College Students at Guangzhou Medical University: 2020.
Health Science and Technology in Guangzhou: 20201A011087.

### Competing Interests
The authors declare that they have no competing interests.

### Author Contributions
- Yukai Dai conceived and designed the experiments, performed the experiments, analyzed the data, prepared figures and/or tables, authored or reviewed drafts of the article, and approved the final draft.
- Qizhong Xu conceived and designed the experiments, performed the experiments, analyzed the data, prepared figures and/or tables, authored or reviewed drafts of the article, and approved the final draft.

- Manqi Xia performed the experiments, prepared figures and/or tables, and approved the final draft.
- Caimin Chen performed the experiments, prepared figures and/or tables, and approved the final draft.
- Xinming Xiong performed the experiments, prepared figures and/or tables, and approved the final draft.
- Xin Yang performed the experiments, prepared figures and/or tables, and approved the final draft.
- Wei Wang conceived and designed the experiments, performed the experiments, prepared figures and/or tables, and approved the final draft.

## Human Ethics

The following information was supplied relating to ethical approvals (*i.e.*, approving body and any reference numbers):

Our study was submitted to the Ethics Committee of the Second Affiliated Hospital of Guangzhou Medical University who approved this study (Clearance certificate number: 2021-ks-KY-07). All patients participating in our study provided informed consents.

## Data Availability

The raw measurements are available in the Supplemental Files. The raw data shows the full-length uncropped pictures for western blot.

## Supplemental Information

Supplemental information for this article can be found online at http://dx.doi.org/10.7717/peerj.17089#supplemental-information.

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
