# Peer review of "Hsa_circ_0001615 downregulation inhibits esophageal cancer development through miR-142-5p/β-catenin"

_PeerJ, doi:10.7717/peerj.17089_

## Round 0.1 · original submission · Major Revisions

Based on the reviewer's reports and my own evaluation, The current article needs to address the raised concerns.

Reviewer 1 ·

Basic reporting

The article maintains a reasonably clear and professional use of English. While it avoids significant ambiguity, there may be occasional instances where further clarity could enhance comprehension.The article adequately references relevant literature, contributing to a foundational understanding of the research. However, there is room for improvement in terms of incorporating a more extensive range of studies and providing even deeper background context.The article adheres to a professional structure, with appropriate sections and the inclusion of figures and tables for data presentation. However, the presentation of these visual aids could be enhanced to improve reader comprehension.The article is somewhat self-contained, with stated research hypotheses and related results. However, there may be opportunities to more explicitly connect the results to the hypotheses and provide additional context to ensure a more comprehensive understanding for readers.

Experimental design

It contains original primary research that aligns reasonably well with the journal's aims and scope but could benefit from a closer alignment. The research question is defined adequately but lacks explicit explanation of its significance within the field. The investigation demonstrates a satisfactory level of rigor, meeting technical and ethical standards to some extent. The methods are described with enough detail to offer a basic understanding, but additional information is needed for complete replication. Overall, the article shows some promise but requires improvement in terms of alignment with the journal's scope, explicitly addressing the research question's significance, and providing more comprehensive methodological details.

Validity of the findings

The article lacks an assessment of its impact and novelty, which are important aspects of research. Understanding the potential significance and originality of the work would enhance its value in the scholarly community.The article encourages meaningful replication, which is a positive aspect. However, it could improve by clearly stating the rationale and benefits of replication to the existing literature, helping readers understand why replication is valuable.The article performs well in providing all underlying data, which are robust, statistically sound, and controlled. This contributes positively to the credibility of the research.

Annotated reviews are not available for download in order to protect the identity of reviewers who chose to remain anonymous.

Reviewer 2 ·

Basic reporting

The article entitled "Downregulation of hsa_circ_0001615 inhibits esophageal cancer process through miR-142-5p/β-catenin." Overall, the manuscript presents valuable insights into the role of hsa_circ_0001615 in inhibiting the progression of esophageal cancer through the miR-142-5p/β-catenin pathway.
o The manuscript maintains a commendable standard of clear and professional English throughout its content.
o The manuscript is well-structured and includes raw data, which enhances its overall quality and credibility.
However, there are a few points that need to be addressed before acceptance:
• While the authors have provided sufficient background information, it would be beneficial to include references to recent publications in the related area to further enrich the context and support the study.
• Line 52: The statement regarding circRNAs lacking 5’-2’ polarity and polyA tails, should be revised to state "5’-3’" for accuracy.
• Full forms of abbreviations need to be provided before their first use in the manuscript for clarity and understanding.
• Maintain consistency in abbreviations in manuscript such as RT-PCR.
• Adhere to the journal's guidelines for reference format.

Experimental design

• Line 140: Ensure citation of K.J. Livak, 2001 for the 2-ΔΔCT method to support the methodology used.
• The loading concentration of protein in western blotting should be explicitly mentioned in the methodology section for reproducibility and clarity of the experimental procedure.
• Fig 3B: It is recommended to clarify the heading on the bar graph for improved comprehension. Using "relative intensity" instead of "relative expression" could enhance the interpretation of results for readers.
• In the first bar graph of Bax in Fig 3B, there appears to be a discrepancy between the reported values of Bax intensity and the observed intensity in the blot image for KYSE-520. Clarification or rectification of this inconsistency is necessary for accurate representation and interpretation of the data.

Validity of the findings

Fig 3B: It is recommended to clarify the heading on the bar graph for improved comprehension. Using "relative intensity" instead of "relative expression" could enhance the interpretation of results for readers.
• In the first bar graph of Bax in Fig 3B, there appears to be a discrepancy between the reported values of Bax intensity and the observed intensity in the blot image for KYSE-520. Clarification or rectification of this inconsistency is necessary for accurate representation and interpretation of the data.

Additional comments

It is recommended to check the typing errors in the manuscript.

---

## Round 0.2 · accepted · Accept

Based on the reviewers' reports and my own evaluation, the current article may be published.

Reviewer 1 ·

Basic reporting

no comment

Experimental design

no comment

Validity of the findings

no comment

Additional comments

The authors addressed all my concerns, the paper is appropriate for publishing.

Reviewer 2 ·

Basic reporting

I would like to express my appreciation for the thorough revisions made by the authors. The changes have significantly improved the clarity and overall quality of the manuscript.

Clarity and Structure:
The revisions have addressed the concerns regarding clarity and structure. The manuscript is now well-organized, and the arguments are presented in a more coherent manner.

Literature Review:
The revised literature review now more effectively connects to the research questions, highlighting the paper's contribution to the existing literature. All relevant and recent studies have been appropriately incorporated.

Experimental design

Methodology:
The additional details provided regarding the methodology have strengthened the rigor of the study. The authors have adequately addressed the limitations, enhancing the credibility of the research.

Validity of the findings

Data Analysis and Results:
The inclusion of visual representations and expanded interpretations significantly enhances the presentation of results. The authors have effectively responded to my earlier suggestions.

Conclusion and Implications:
The revised conclusion now effectively summarizes key findings and discusses their broader implications. The inclusion of potential future research directions is appreciated.

Additional comments

References:
The authors have addressed the issues related to references, ensuring consistency in citation format and verifying accuracy.
Overall Evaluation:
Considering the authors' diligent revisions, I recommend accepting the manuscript for publication in Peer J. The paper now meets the standards of quality and originality expected by the journal.